# LTR Retroelements and Bird Adaptation to Arid Environments

**DOI:** 10.3390/ijms24076332

**Published:** 2023-03-28

**Authors:** Elisa Carotti, Edith Tittarelli, Adriana Canapa, Maria Assunta Biscotti, Federica Carducci, Marco Barucca

**Affiliations:** Dipartimento di Scienze della Vita e dell’Ambiente, Università Politecnica delle Marche, Via Brecce Bianche, 60131 Ancona, Italy

**Keywords:** transposable elements, genome evolution, vertebrates, Aves, molecular adaptation, Australian bird evolution

## Abstract

TEs are known to be among the main drivers in genome evolution, leading to the generation of evolutionary advantages that favor the success of organisms. The aim of this work was to investigate the TE landscape in bird genomes to look for a possible relationship between the amount of specific TE types and environmental changes that characterized the Oligocene era in Australia. Therefore, the mobilome of 29 bird species, belonging to a total of 11 orders, was analyzed. Our results confirmed that LINE retroelements are not predominant in all species of this evolutionary lineage and highlighted an LTR retroelement dominance in species with an Australian-related evolutionary history. The bird LTR retroelement expansion might have happened in response to the Earth’s dramatic climate changes that occurred about 30 Mya, followed by a progressive aridification across most of Australian landmasses. Therefore, in birds, LTR retroelement burst might have represented an evolutionary advantage in the adaptation to arid/drought environments.

## 1. Introduction

Transposable elements (TEs) are DNA sequences capable of replicating, moving, and integrating into new regions of the genome [1]. Two major classes can be identified according to their mode of transposition: retrotransposons (class I) and DNA transposons (class II). Class I TEs are able to propagate their copy sequences in the host genome by reverse transcription of an RNA intermediate molecule through a copy-and-paste mechanism. Elements of class I include long terminal repeat (LTR) retrotransposons and non-LTR retrotransposons. The latter encompass long interspersed nuclear elements (LINEs) and short interspersed nuclear elements (SINEs). Class II elements are generally removed from their original location in order to be inserted in a different portion of the genome by a cut-and-paste mechanism [1].

TEs play a key role in the response of genome to external stimuli, being activated by biotic and abiotic factors. Their activation might be due to impaired functionality of TE silencing mechanisms (e.g., RNA interference by piRNAs and/or DNA methylation), resulting in the consequent acceleration of genetic variability and genome reshaping, leading species to adapt rapidly to new conditions [2]. Therefore, TE activity-mediated response of species to environmental stressors such as variations in temperature, humidity, and salinity could facilitate adaptation to climate changes and species radiation [3,4,5,6,7,8,9,10]. This dynamic interplay between TEs and the environment has been well reported for plant and *Drosophila* [11,12,13], while this issue has only recently been covered in vertebrates [14,15,16,17].

Birds have small genome sizes (0.89 Gb in the black-chinned hummingbird to 2.11 Gb in the ostrich) [18,19,20], similar to other flying organisms (bats and pterosaurs) [21,22,23] that have convergently evolved constricted genome sizes [22]. Their genomes are known to contain lower levels of TEs (~4 to 10%) [24], with the exception of woodpeckers in which they make up 20% [20,25,26,27]. Moreover, bird genomes are characterized by a low diversity of TE superfamilies mainly belonging to LINEs, followed by LTR retroelements, DNA transposons, and SINEs. With 39–88% of all TE copies, chicken repeat 1 (CR1) elements, belonging to non-LTR retroelements, are the dominant TE subfamily in birds. The long-term stability in genome size observed in these endothermic organisms is due to the balance between the rate of DNA gain through transposition and that of DNA loss, the so-called “accordion” model of genome evolution [20].

The modern birds, or Neornithes, are a monophyletic group including Palaeognathae and Neognathae [28], originating from the Palaeocene (65 Mya) after the Cretaceous-Paleogene (K-Pg) transition [29]. Australia has played a key role in bird radiation as most extant birds can reconduct their evolutionary history to this continent [30]. It has always been considered the landmass with the largest and most fascinating biodiversity due to its long period of isolation after separation from Antarctica, estimated in the late Eocene (35 Mya) [31]. Significant ecological, geological, and climatic events have contributed to the current subdivision of the Australian continent into different biomes that are currently classified according to the Köppen-Geiger climate type system in three main climates: tropical (8.3%), arid (77.8%), and temperate (13.9%) [32]. These features have contributed to the development of the surprising Australian biology characterized by flora and fauna, having unique characteristics and a high level of endemism on a regional and continental scale. Moreover, Australia presents an interesting biogeography of species whose distribution extends from temperate to tropical zones [33].

The aim of this work was to investigate if a correlation exists between the TE type content in bird genomes and the environmental changes occurred in Australia during Oligocene (33 Mya). Therefore, the TE fraction of 29 bird species belonging to a total of 11 orders was analyzed through a bioinformatic approach (Figure 1 and Table 1).

## 2. Results

The genomes of seven Australian birds, along with 22 species living in other geographical areas, belonging to a total of 11 orders, were masked to investigate the composition of their repetitive fraction. The total TE percentage was lower than 10 for most of the analyzed species. It was noteworthy that the TE quantity identified in the two species belonging to Piciformes exceeded 25% (Figure 2). In Figure 3, the relative abundance of each TE type was reported. Overall, the major impact was represented by LINE retroelements, including four Australian species. Interestingly, for the other three species living in Australia, namely *M. undulatus*, *T. guttata*, and *D. novaehollandiae*, LTR retroelements showed the highest relative abundance. This finding was shared also by *M. monachus,* distributed in South America, and *H. rustica,* having a worldwide distribution. Moreover, patterns evidenced by TE relative abundance analysis were not consistent with species taxonomy (Figure 3), as also supported by the variation partitioning analysis (Figure 4). For 11 species, chromosome-level genome assemblies were not present in public repositories. In these cases, the masking analysis of genome repetitive fraction was performed on available scaffolds. In order to verify a possible bias due to assembly quality, comparisons of TE content were carried out on 14 species for which both scaffold and chromosome data were available (Appendix A). In general, the percentage of total TEs obtained for scaffolds was on average underestimated by 17.8%. Concerning the two most abundant TE types (LINE and LTR retroelements), compared to total TEs, LINE retroelements showed an average variation of 6.6%, while LTR retroelements showed 4.8%. In the latter, the variation always represented an underestimation of LTR content obtained by analyzing scaffold assemblies. Therefore, the extent of these discrepancies suggested that patterns of TE type detected for species with only available scaffolds are likely to be reliable. In *D. novaehollandiae*, the difference in TE content obtained by analyzing the chromosome and scaffold assemblies was high, suggesting low reliability of the scaffold-level genome assembly. Hence, data relating to this species were not considered. In the latter, about 68% of TEs were not masked, probably due to low assembly quality of regions harboring these repetitive elements.

Genomic TE sequence divergence was analyzed using Kimura distance method (Appendix A). In Figure 5, Kimura landscapes were reported for Passeriformes (passerines), Psittaciformes (parrots), and Palaeognathae (ratites and tinamous), taxa that have species with an abundance of LTR retroelements.

In the case of Passeriformes (Figure 5, panel A), all the considered species showed amplification bursts of LTR retroelements. However, *P. domesticus* and *F. albicollis* showed an intense expansion of LINE retroelements, leading the latter TE type to be the most abundant in their genomes. Additionally, *T. guttata* experienced a LINE burst, although LTR retroelements remained the most prevalent TE type. Concerning Psittaciformes, *M. undulatus* and *M. monachus* presented TE amplifications dominated by LTR retroelements. Despite a recent LTR burst, the *A. aestiva* genome showed a prevalence of LINE retroelements due to an amplification of these mobile elements, as revealed by the Kimura landscape (Figure 5, panel B). Bursts characterized by a dominance of LTR retroelements were not recorded for *S. habroptila*. Among Palaeognathae, *D. novaehollandiae* was the only species showing a remarkable LTR amplification burst (Figure 5, panel C). Moreover, *T. guttata*, *M. undulatus*, and *D. novaehollandiae* landscapes showed recent and active DNA transposon elements.

## 3. Discussion

In this paper, we investigated the TE landscape in bird genomes to understand if a correlation exists between the amount of specific TE types and environmental changes that characterized the Oligocene era in Australia. Our results confirmed the idea that LINE retroelements are not predominant in all species of this evolutionary lineage [20,34]. Indeed, five of the 29 analyzed species showed a dominance of LTR retroelements, and interestingly, all of them belong to Australian species or to taxa with an Australian ancestor. *M. undulatus*, *T. guttata*, and *D. novaehollandiae* are widely distributed in the Australian landmass. *M. monachus* lives in Eastern South America; however, similar to *M. undulatus*, it belongs to parrots that diversified from an Australian ancestor [35]. Our analyses showed the presence of a high LTR retroelement amount in the genome of a fifth species, *H. rustica*. This species has a worldwide distribution but, similar to *T. guttata*, it belongs to Passeriformes, an order that also had an Australian origin [30]. Moreover, the ancestors of these species arose in the same time window, about 30 Mya in Oligocene. During the transition from Eocene to Oligocene, the Earth was affected by strong climate changes leading to a dramatic global cooling that determined mass extinction as well as species diversification [36,37]. In the Australian continent, these events caused the progressive rainforest reduction and an increase of aridification. Although some species went extinct, other taxa increased diversification rates and adapted to new niches [36]. Since TEs can be activated in response to environmental stress conditions [3,8,14,15,16], the amplification of LTR retroelements in the species analyzed here might have represented an evolutionary advantage (Figure 6).

Phylogenetic studies [38,39] have highlighted a large evolutionary distance between the taxa exhibiting an LTR retroelement prevalence, leading to the hypothesis that the expansion of these TEs occurred independently in bird lineages. This observation is also supported by variation partitioning analysis, which revealed a better correlation between LTR retroelement content and climate changes that occurred in Australia and Antarctica during the Eocene-Oligocene transition than the phylogeny of the species analyzed. In plants, LTR retroelements have been reported to be activated under drought stress conditions [40,41,42], and in animals, a prevalence of these TE elements has been identified in the genome of the genus *Camelus* adapted to arid environments [43]. In support of our hypothesis, Australian species living in temperate and tropical regions, such as *P. strigoides*, *A. lathami*, and *P. torquatus* (Figure 7), showed a limited fraction of LTR retroelements and a strong prevalence of LINE retroelements, as reported also for the *C. casuarius* genome. This species is strictly related to *D. novaehollandiae,* and they diverged from a common ancestor during Oligocene [44]. It is distributed in New Guinea and in York Peninsula (far North Queensland, Australia), which agrees with Moyle et al. (2016) [30] that proposed New Guinea as a refuge for relictual lineages, restricted to the Australian northern continental margin after the progression of aridification (Figure 7).

Other species belonging to Palaeognathae, such as *S. camelus australis* and *R. americana*, which diverged from Casuariiformes (*C. casuarius* and *D. novaehollandiae*) before the Oligocene, also showed a prevalence of LINE retroelement. This finding corroborates the hypothesis that the LTR retroelement expansion was specific to bird ancestors that lived in Australia during this geological era. Regarding parrots, the LINE retroelement dominance in *S. habroptila*, an endemic species of New Zealand, might be explained by the early divergence of its ancestor from that of the other analyzed parrots, which occurred during Eocene [45]. Moreover, New Zealand was not affected by the remarkable climate changes that characterized Australia in the Oligocene (Figure 7). Concerning the two parrots living in South America, *A. aestiva* mobilome showed a prevalence of LINE retroelements, while LTR retroelements were dominant in *M. monachus*. Their ancestors migrated to Antarctica before the separation of this continent from Australia, which was dated around 55–45 Mya [46]. Subsequently, the ancestor of *Amazona* evolutionary lineage probably colonized South America before that of *Myiopsitta* (Figure 6). The diploid chromosome number of 2 n = 48 described for *M. monachus* represents an exception among the South American Psittacidae (which are generally 2 n = 70) [47]. This atypical karyotype might support the hypothesis of a diverse origin for this species. The LTR retroelement expansion observed in *Myiopsitta* might have occurred in its ancestor, which lived in Antarctica, in response to the aridification that characterized this landmass in the Oligocene. Moreover, this parrot retained an LTR retroelement abundance that might have favored its current distribution in arid areas. In contrast, the *A. aestiva* ancestor, which was already present in South America, was not affected by the climate stress conditions that were happening in the Antarctic and Australian landmasses starting from the Oligocene. Alternatively, if *A. aestiva* and *M. monachus* originated from a common ancestor that migrated from Antarctica to South America, their different TE composition might be explained by considering the recurrent chromosome reshuffling events that are typical of parrot genomes [48].

In the case of Passeriformes, the analyzed species had a common ancestor that lived in Australia during the Oligocene [35]. Indeed, it is known that passerines dispersed from Australia via South-East Asia ca. 23 Mya [30]. This observation could explain the LTR retroelement bursts observed in the Kimura landscapes of species analyzed here. However, only *T. guttata* and *H. rustica* maintained a prevalence of LTR retroelements: for *T. guttata* this condition might be due to its wide distribution in Australia and mainly in arid environments; for *H. rustica* this feature might have been at the base of its wide adaptability to different environments, which led to its current worldwide distribution. The LTR retroelement decrease observed in the other analyzed Passeriformes might be the result of a balance between the rate of DNA gains through transposition and that of DNA loss, as proposed in the “accordion” model of bird genome evolution [20].

Interestingly, the species *T. guttata*, *M. undulatus*, and *D. novaehollandiae* presented active DNA transposons. It is known that in vertebrates, these elements are usually defective and thus not active. Indeed, according to the evolutionary life-cycle of DNA transposons, after host genome invasion, these elements undergo bursts of amplification. In this phase, DNA transposons are prone to accumulate mutations that make them inactive. This process is called vertical inactivation [49,50]. Therefore, the DNA transposons identified in these three bird species might be in the initial phase of this cycle. The presence of DNA transposon active copies even in other bird species not correlated with Australia (Appendix A) suggests that this trait is not linked to climate events that occurred from Oligocene in this landmass.

## 4. Materials and Methods

In this study, 29 bird species belonging to 11 orders were considered (Table 1). Unmasked genomes of these species were obtained from the public database NCBI GenBank (https: //www.ncbi.nlm.nih.gov/genome/, accessed on 13 January 2022) and ENSEMBL (https://www.ensembl.org/index.html, accessed on 13 January 2022). Accession numbers and genome assembly levels were reported in Appendix A. To identify TEs in the considered species and create the species-specific de novo TE libraries, we followed the pipeline developed by Carotti et al., 2021 [16]. Using RepeatScout v 1.0.6 [51], de novo TEs were identified: the “build_lmer_table” module generated a table of lmer frequency and “RepeatScout” extracted the repeats that were then filtered with “filter-stage-1.prl” script in order to remove low complexity sequences. This step allows generating the “build_lmer_table” that was then used by RepeatMasker v 4.1.0 (http://www.repeatmasker.org/cgi-bin/WEBRepeatMasker, accessed on 13 January 2022) as a library to extract the repeat sequences. Those repeats accounted for less than 10 times were removed by RepeatScout “filter-stage-2.prl” script. Additional filtering steps were performed to remove sequences that were not identified as TEs. Briefly, applying a threshold e-value of 1 × 10^−50^, BLASTX [52] search against the Uniprot–Swissprot database [53] and Interproscan v5.34–73.0 [54] were performed. Among discarded elements, possible domesticated transposons filtered out in the previous step were searched by HMMER [55]. Sequences containing domains ascribable to integrase, reverse transcriptase, and transposase with e-value lower than 1 × 10^−5^ were reintegrated in the TE library. As the last step, the remaining sequences were classified using TEclass-2.13 (https://www.bioinformatics.uni-muenster.de/tools/teclass/index.hbi?, accessed on 13 January 2022) to obtain the specie-specific TE libraries. These were used by RepeatMasker to mask each genome specifying the “-a” argument in the command line. In addition, the scripts “calcDivergenceFromAlign.pl” and “createRepeatLandscape.pl”, included in the RepeatMasker package, were applied to estimate TE divergence and transposition history in bird genomes. Kimura distances (rate of transitions and transversions) were calculated between TE sequences identified in the genome and TE consensus from the library.

To evaluate whether the LTR retroelement predominance observed in some of the analyzed species was related to climate changes that occurred in Australia and Antarctica during the Eocene-Oligocene transition (explanatory variable X1) or to phylogenetic relationships (explanatory variable X2), a variation partitioning analysis was performed using the vegan package 2.5.5 [56]. For each pair of the 28 species considered in the analysis, both explanatory variables were compared with data related to the difference between the LTR retroelement relative abundances (assigned as response variable X). To set the explanatory variable X1, *D. novaehollandiae* (Casuariiformes), *F. albicollis*, *H. rustica*, *P. domesticus*, and *T. guttata* (Passeriformes), and *A. aestiva*, *M. monachus*, and *M. undulatus* (Psittaciformes) were considered as species showing an ancestor that probably experienced an LTR expansion as a consequence of the aforementioned climate changes. For the explanatory variable X2, phylogenetic distances were evaluated using the 16S rDNA *p*-distance matrix (Appendix A). Note that *C. europaeus* was not included in the analysis due to the lack of 16S rDNA sequence in public databases.

## 5. Conclusions

Our results could constitute further evidence that TEs have been active players in bird genome evolution, favoring their adaptation. Although the major part of the bird mobilome is constituted by LINE retroelements, our findings highlighted an LTR retroelement dominance in species with an Australian-related evolutionary history. Indeed, these lineages underwent a radiation about 30 Mya when Earth’s climate was destabilized by dramatic changes, and Australia started to experience progressive aridification across most of its landmasses. The bird LTR retroelement expansion might have represented an evolutionary advantage in the adaptation to arid/drought environments, in line with that reported in other species [40,41,42,43]. It is known that LTR retroelements can control the transcription of host gene by inserting cis-regulatory sequences such as enhancers, promoters, and insulators [57,58]. Therefore, these mobile elements can modify the transcriptional activity of neighboring host genes leading to short-term adaptation of physiological functions in response to environmental changes and long-term evolutionary changes [58]. Further investigations will be useful to explore through what cis regulatory activity LTR retroelements might have contributed to the adaptability of birds to environmental changes.

## Figures and Tables

**Figure 1 ijms-24-06332-f001:**
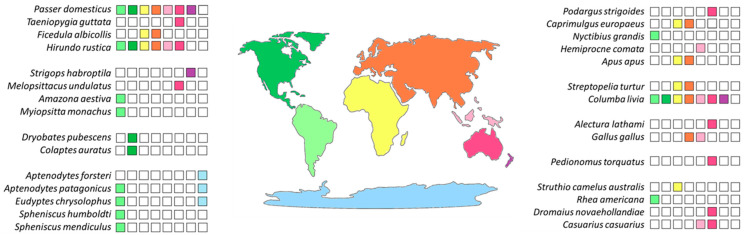
Species considered in this study and their geographical distribution are shown. The landmasses are colored as follow: North and Central America in dark green; South America in light green; Africa in yellow; Antarctica in blue; Europe-Asia in orange; Australia in magenta; Indonesia and New Guinea in pink; New Zealand in purple. The colored square boxes indicate the geographical distribution of each species analyzed in this study.

**Figure 2 ijms-24-06332-f002:**
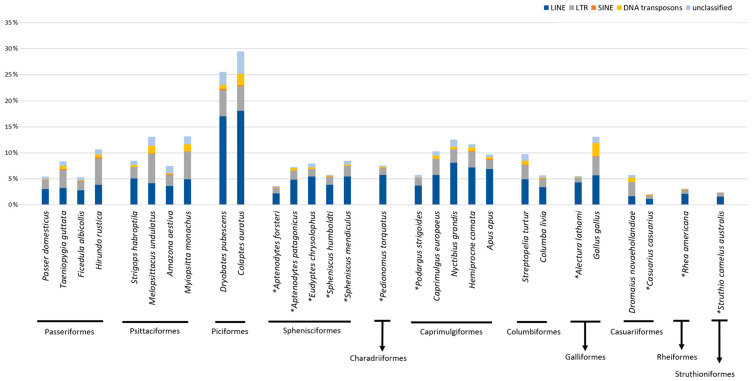
The percentage of total transposable elements (TEs) masked in the genomes of the studied species. The histogram displays the percentage of the main TE types. * indicates species for which the analysis was performed on scaffold-level genome assembly.

**Figure 3 ijms-24-06332-f003:**
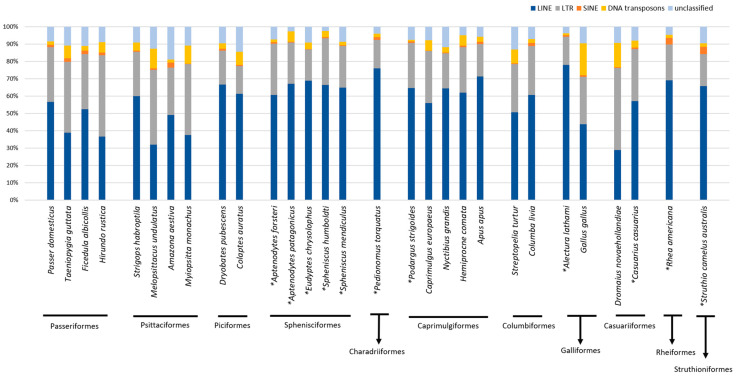
Relative abundance of TE types in the mobilome of the bird species analyzed. * indicates species for which the analysis was performed on scaffold-level genome assembly.

**Figure 4 ijms-24-06332-f004:**
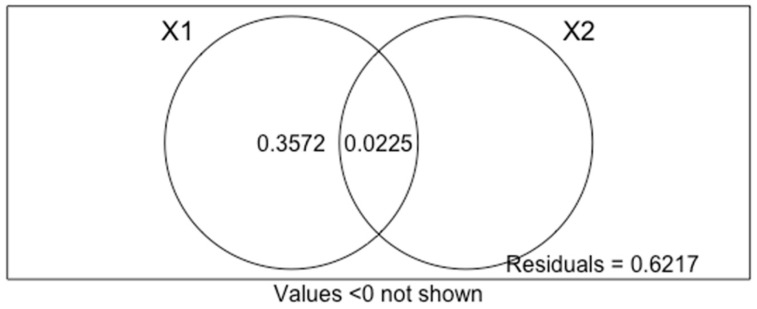
Venn diagrams obtained by the variation partitioning analysis (VPA) using redundancy analysis (RDA). The partition of the variation of a response variable (X) between two sets of explanatory variables (X1 and X2) is shown. Each circle represents the portion of variation accounting for each explanatory variable or a combination of the explanatory matrices. The intersection between the two circles represents the amount of variation explained by both variables X1 and X2. The response variable (X) is the difference between the LTR retroelements relative abundances; the explanatory variable X1 represents species that have an ancestor that probably experienced an LTR expansion as consequence of climate changes that occurred in Australia and Antarctica during the Eocene-Oligocene transition; the explanatory variable X2 indicates the sequence divergence obtained by p-distance matrix using 16S rDNA sequences.

**Figure 5 ijms-24-06332-f005:**
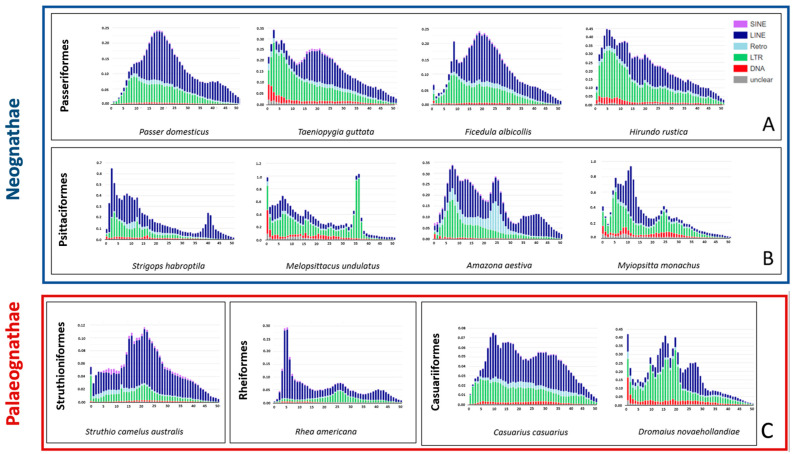
TE landscape plots obtained using Kimura distance-based copy divergence analyses of some species belonging to Neognathae (**blue box**) and Palaeognathae (**red box**). X axis: Kimura substitution level (CpG adjusted); Y axis: percent of genome. Panel (**A**) includes Passeriformes; panel (**B**) includes Psittaciformes; panel (**C**) includes Struthioniformes, Rheiformes, and Casuariformes.

**Figure 6 ijms-24-06332-f006:**
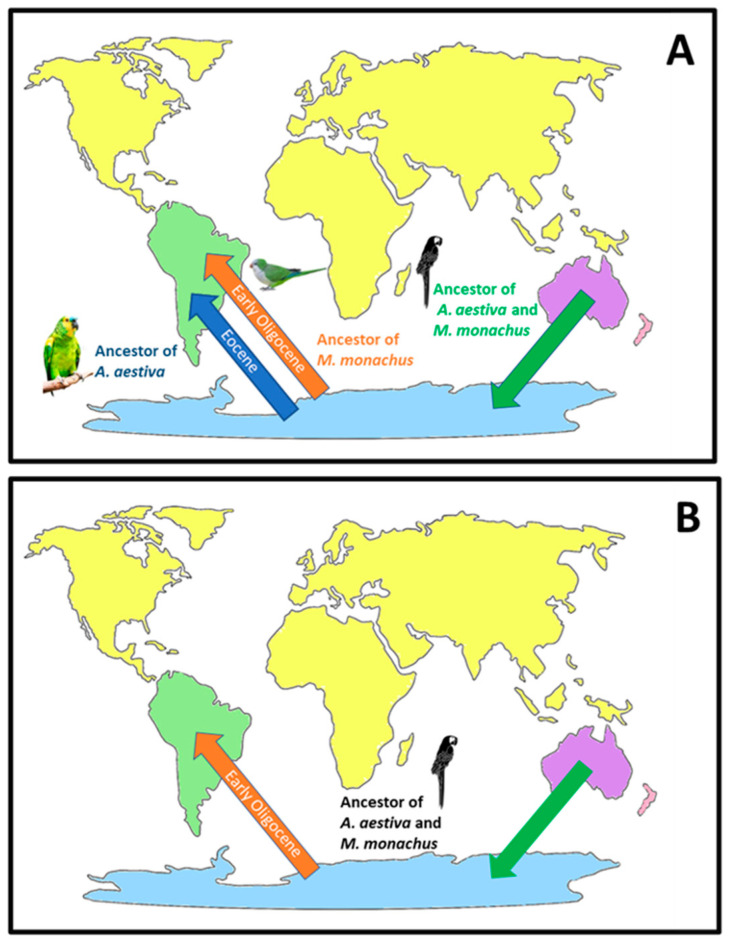
Hypotheses suggested for *A. aestiva* and *M. monachus* evolutionary radiation. Panel (**A**): the ancestors of *A. aestiva* and *M. monachus* migrated from Australia to Antarctica (green arrow). The ancestor of *A. aestiva* migrated from Antarctica to South America before the dramatic cooling event occurred during the Eocene-Oligocene transition that led to the progressive aridification of Antarctica and Australian landmasses (blue arrow). The ancestor of *M. monachus* migrated from Antarctica to South America after this climate change event (orange arrow). Panel (**B**): the ancestors of *A. aestiva* and *M. monachus* migrated from Australia to Antarctica (green arrow) and then to South America after dramatic climate change events occurred about 30 Mya (orange arrow). Please see the discussion section for further details. For convenience, the current distribution of continents is shown.

**Figure 7 ijms-24-06332-f007:**
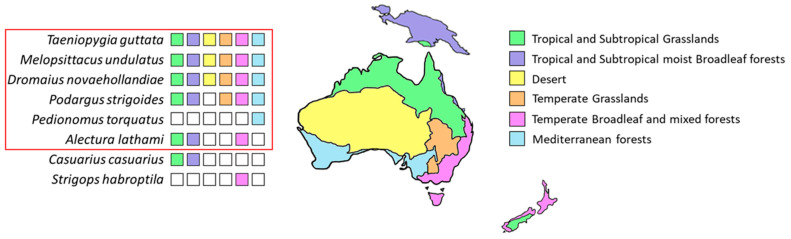
On the left side biome-related distribution is reported for eight species considered in this study. The red box groups the Australian species; *C. casuarius* is distributed in New Guinea and in York Peninsula (far North Queensland, Australia); *S. habroptila* is an endemic species of New Zealand. In the geographical map the biomes present in these areas are colored according to the legend.

**Table 1 ijms-24-06332-t001:** Species analyzed in this study.

Superorder	Order	Species
Neognathae	Caprimulgiformes	*Apus apus*
		*Caprimulgus europaeus*
		*Hemiprocne comata*
		*Nyctibius grandis*
		*Podargus strigoides*
	Charadriiformes	*Pedionomus torquatus*
	Columbiformes	*Columba livia*
		*Streptopelia turtur*
	Galliformes	*Alectura lathami*
		*Gallus gallus*
	Passeriformes	*Ficedula albicollis*
		*Hirundo rustica*
		*Passer domesticus*
		*Taeniopygia guttata*
	Piciformes	*Colaptes auratus*
		*Dryobates pubescens*
	Psittaciformes	*Amazona aestiva*
		*Melopsittacus undulatus*
		*Myiopsitta monachus*
		*Strigops habroptila*
	Sphenisciformes	*Aptenodytes forsteri*
		*Aptenodytes patagonicus*
		*Eudyptes chrysolophus*
		*Spheniscus humboldti*
		*Spheniscus mendiculus*
Palaeognathae	Casuariiformes	*Casuarius casuarius*
		*Dromaius novaehollandiae*
	Rheiformes	*Rhea americana*
	Struthioniformes	*Struthio camelus australis*

## Data Availability

Accession numbers of analyzed data are contained in Appendix A.

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
