# Peer review of "LTR Retroelements and Bird Adaptation to Arid Environments"

_ijms, 2023, doi:10.3390/ijms24076332_

Round 1

Reviewer 1 Report

In this manuscript, the genome of 29 species of birds were mined in order to define their TE repertoire. The main finding is that LINEs are not always predominant in these genomes, Interestingly, the Authors link the observed TE expansion with the climate changes occurred during the Earth history. The manuscript is overall well-written and the conclusions are well supported. 

I have some comments that will hopefully improve the discussion and the readers' interest.

The main point of discussion is on the abundance of LTR retrotransposons in some of the analyzed species. While this is a relevant point raised by the Authors, some other observations can be mede. 

If I am not wrong, DNA elements are well represented in some of the genomes analyzed and these elements are also young in the genome of at least three species (T. guttata, M. undulatus, D. novaehollandiae), as judged from the Kimura distance-based divergence plots. In my opinion, this could be a relevant point of discussion, since active DNA elements are usually poorly represented in vertebrate genomes (10.1007/s00018-004-4232-7). 

The authors do not discuss on the potential effect of transposition burst that may have occurred in response of climate changes. It is widely acknowledged that TEs are potent source of structural and transcriptional plasticity and variability in many organisms (see doi:10.1098/rstb.2019.0347 and doi: 10.3390/biology9020025).

It is not clear from the manuscript if the authors have annotated the mobile elements of the species examined. Anyway, studies like this one proposed in the manuscript could be useful to track the structure of populations, especially this that are at risk of extinction or ecologically relevant. The Batzer group have already demonstrated that the definition of a lineage specific insertions (10.1093/gbe/evx130) can be useful to define the population's structure (10.1093/gbe/evx184) and for the phylogenetic reconstruction (10.1186/s13100-018-0118-3, 10.1186/s13100-019-0187-y).

Author Response

In this manuscript, the genome of 29 species of birds were mined in order to define their TE repertoire. The main finding is that LINEs are not always predominant in these genomes, Interestingly, the Authors link the observed TE expansion with the climate changes occurred during the Earth history. The manuscript is overall well-written and the conclusions are well supported. 

I have some comments that will hopefully improve the discussion and the readers' interest.

The main point of discussion is on the abundance of LTR retrotransposons in some of the analyzed species. While this is a relevant point raised by the Authors, some other observations can be mede. 

If I am not wrong, DNA elements are well represented in some of the genomes analyzed and these elements are also young in the genome of at least three species (T. guttata, M. undulatus, D. novaehollandiae), as judged from the Kimura distance-based divergence plots. In my opinion, this could be a relevant point of discussion, since active DNA elements are usually poorly represented in vertebrate genomes (10.1007/s00018-004-4232-7).

Answer: thanks to the reviewer for his/her comments. This interesting aspect has been added both in the result and discussion sections.

The authors do not discuss on the potential effect of transposition burst that may have occurred in response of climate changes. It is widely acknowledged that TEs are potent source of structural and transcriptional plasticity and variability in many organisms (see doi:10.1098/rstb.2019.0347 and doi: 10.3390/biology9020025).

Answer: thanks to the reviewer for his/her comments. We have added this issue in the conclusion section to better explain through which mechanisms LTR retroelements might have contributed to bird adaptability.

It is not clear from the manuscript if the authors have annotated the mobile elements of the species examined. Anyway, studies like this one proposed in the manuscript could be useful to track the structure of populations, especially this that are at risk of extinction or ecologically relevant. The Batzer group have already demonstrated that the definition of a lineage specific insertions (10.1093/gbe/evx130) can be useful to define the population's structure (10.1093/gbe/evx184) and for the phylogenetic reconstruction (10.1186/s13100-018-0118-3, 10.1186/s13100-019-0187-y).

Answer: we thank the reviewer for his/her comment and for the papers suggested. In our work we did not annotated the TEs but we classified them using TEclass in the main TE types (DNA transpson, LINE, SINE and LTR).

Reviewer 2 Report

    Your aim was to investigate the TE landscape in bird genomes to understand if a correlation exists between the amount of specific TE types and environmental changes that characterized the Oligocene era in Australia. Your results  highlighted an LTR retroelement dominance in species with an Australian-related evolutionary history.  However, I don't think that your hypothesis were well supported by your current results. Please afford necessary Results, Table and Figures. Also, language need to be edited

     I have highlighted and noted more detailed comments in the main text. 

Author Response

Comments and Suggestions for Authors

    Your aim was to investigate the TE landscape in bird genomes to understand if a correlation exists between the amount of specific TE types and environmental changes that characterized the Oligocene era in Australia. Your results highlighted an LTR retroelement dominance in species with an Australian-related evolutionary history.  However, I don't think that your hypothesis were well supported by your current results. Please afford necessary Results, Table and Figures. Also, language need to be edited

     I have highlighted and noted more detailed comments in the main text. 

Answer: we thank the reviewer for his/her suggestions. We have replied all comments in the pdf file.  Moreover, to provide a major support to our results we moved the Figure S1 as main figure now named Figure 4 reporting the results of variation partitioning analysis. In Supplementary Material we added information about this analysis in particular a table in which we reported the accession numbers of rDNA 16S sequences (Supplementary Table S3) and the p-distance matrix (Supplementary Table S4). Accordingly, the text was modified in the Discussion section. We also clarified the aspect related to phylogeny adding the citations of works about phylogenetic studies performed on bird species.

Reviewer 3 Report

The article brings important data on the adaptive contribution of TEs in the evolution of the bird genome, it is well written and discussed corroborating the relevance of studies on the mobiloma not only in birds, but also in other taxa. As a contribution, I suggest to edit the Supplementary Figure S2 for the same formatting of Figures 2 and 3. Ordereding the graph bars with the name of the species and perhaps abbreviating the genus.

Author Response

The article brings important data on the adaptive contribution of TEs in the evolution of the bird genome, it is well written and discussed corroborating the relevance of studies on the mobiloma not only in birds, but also in other taxa. As a contribution, I suggest to edit the Supplementary Figure S2 for the same formatting of Figures 2 and 3. Ordereding the graph bars with the name of the species and perhaps abbreviating the genus.

Answer: thank the reviewer for this comment. The figure has been modified following the suggestions.

Round 2

Reviewer 1 Report

The Authors have made major changes according to my previous suggestions. I am sorry that the Authors haven't discussed the useful application of TEs for population studies and analyses. This would add value to the study and possibly open additional points of discussion in the scientific community (even if the annotation has not been already performed in this study).

In conclusion, I think that the manuscript can be accepted for publication, although I encourage the Authors to include the above-suggested point of discussion.

Please, note that several grammar errors and typos are still present in the text and that should be fixed (some examples are reported below). In addition I would refer to "putatively active elements" throughout the text, since there is no demonstration (either direct or indirect) of their activity.

l269 "high distance" -> "large evolutionary distance"

l377 "no active"

l382 "no correlated"

Author Response

Open Review

Quality of English Language

( ) English very difficult to understand/incomprehensible
( ) Extensive editing of English language and style required
( ) Moderate English changes required
(x) English language and style are fine/minor spell check required
( ) I am not qualified to assess the quality of English in this paper

Comments and Suggestions for Authors

The Authors have made major changes according to my previous suggestions. I am sorry that the Authors haven't discussed the useful application of TEs for population studies and analyses. This would add value to the study and possibly open additional points of discussion in the scientific community (even if the annotation has not been already performed in this study).

In conclusion, I think that the manuscript can be accepted for publication, although I encourage the Authors to include the above-suggested point of discussion.

Please, note that several grammar errors and typos are still present in the text and that should be fixed (some examples are reported below). In addition I would refer to "putatively active elements" throughout the text, since there is no demonstration (either direct or indirect) of their activity.

l269 "high distance" -> "large evolutionary distance"

l377 "no active"

l382 "no correlated"

Answer: thanks for considering acceptable our manuscript. We agree about the potential use of TEs for population studies and analyses. However, our paper does not cover this issue.